# Henipavirus Matrix Protein Employs a Non-Classical Nuclear Localization Signal Binding Mechanism

**DOI:** 10.3390/v15061302

**Published:** 2023-05-31

**Authors:** Camilla M. Donnelly, Olivia A. Vogel, Megan R. Edwards, Paige E. Taylor, Justin A. Roby, Jade K. Forwood, Christopher F. Basler

**Affiliations:** 1School of Dentistry and Medical Sciences, Charles Sturt University, Wagga Wagga, NSW 2678, Australia; cdonnelly@csu.edu.au (C.M.D.); paige.taylor17@outlook.com (P.E.T.); jroby@csu.edu.au (J.A.R.); 2Department of Microbiology, Icahn School of Medicine at Mount Sinai, New York, NY 10029, USA; olivia.vogel@mssm.edu; 3Institute for Biomedical Sciences, Georgia State University, Atlanta, GA 30303, USA; medwar02@student.ubc.ca; 4School of Population and Public Health, Faculty of Medicine, The University of British Columbia, Vancouver, BC V6T 1Z3, Canada

**Keywords:** Nipah virus, Hendra virus, henipavirus, matrix protein, importin, X-ray crystallography

## Abstract

Nipah virus (NiV) and Hendra virus (HeV) are highly pathogenic species from the *Henipavirus* genus within the paramyxovirus family and are harbored by *Pteropus* Flying Fox species. Henipaviruses cause severe respiratory disease, neural symptoms, and encephalitis in various animals and humans, with human mortality rates exceeding 70% in some NiV outbreaks. The henipavirus matrix protein (M), which drives viral assembly and budding of the virion, also performs non-structural functions as a type I interferon antagonist. Interestingly, M also undergoes nuclear trafficking that mediates critical monoubiquitination for downstream cell sorting, membrane association, and budding processes. Based on the NiV and HeV M X-ray crystal structures and cell-based assays, M possesses a putative monopartite nuclear localization signal (NLS) (residues ^82^KRKKIR^87^; NLS1 HeV), positioned on an exposed flexible loop and typical of how many NLSs bind importin alpha (IMPα), and a putative bipartite NLS (^244^RR-10X-KRK^258^; NLS2 HeV), positioned within an α-helix that is far less typical. Here, we employed X-ray crystallography to determine the binding interface of these M NLSs and IMPα. The interaction of both NLS peptides with IMPα was established, with NLS1 binding the IMPα major binding site, and NLS2 binding as a non-classical NLS to the minor site. Co-immunoprecipitation (co-IP) and immunofluorescence assays (IFA) confirm the critical role of NLS2, and specifically K258. Additionally, localization studies demonstrated a supportive role for NLS1 in M nuclear localization. These studies provide additional insight into the critical mechanisms of M nucleocytoplasmic transport, the study of which can provide a greater understanding of viral pathogenesis and uncover a potential target for novel therapeutics for henipaviral diseases.

## 1. Introduction

Henipaviruses include highly lethal, zoonotic, paramyxoviruses that harbor a single-stranded RNA negative sense genome of approximately 18 kb. The two prototypical henipavirus species are Nipah virus (NiV) and Hendra virus (HeV). They are pleiomorphic to spherical in shape with genomes comprised of six genes that encode six structural proteins, nucleocapsid (N), phosphoprotein (P), matrix (M), fusion (F), attachment (G), and large (L), and three accessory proteins C, V, and W [1,2].

Henipaviruses first emerged in a 1994 outbreak of HeV when a spill-over occurred from the reservoir host, pteropid bats, into racehorses in the Brisbane suburb of Hendra, QLD, Australia. Subsequent human infections occurred due to close contact with the infected horses. Symptoms include acute respiratory distress, neurological symptoms, and encephalitis [3]. Since the 1994 outbreak, 88 confirmed equine cases and more than 20 unresolved cases have resulted in death or euthanasia, with the most recent spill-over occurring in Mackay, QLD, Australia, in July 2022 [4]. Additionally, seven human cases of HeV have occurred, and a new strain of HeV, Hendra virus genotype 2 (HeV-g2), has since been characterized [4,5,6].

Contrary to HeV infections, human-to-human transmission can occur with NiV [7]. The initial outbreak of NiV occurred in 1998 when a spill-over event infected a piggery, resulting in 256 human cases, 105 deaths, and over 1 million pigs slaughtered to contain the disease [8,9,10]. The second strain of NiV emerged in Bangladesh and India in 2001, with human infection resulting from direct contact with fruit contaminated with bat secretions [11]. More recent NiV outbreaks have been reported, including in September 2021, with associated case fatality rates as high as 70–100% [12,13,14].

In addition to HeV and NiV, the characterization of several other henipaviruses has occurred. *Ghanaian bat henipavirus*, *Cedar henipavirus,* and *Angavokely henipavirus* were detected in bat species, whereas *Mojiang henipavirus* specimens were collected from yellow-breasted rats [15]. A likely zoonosis from shrews, *Langya henipavirus,* was recently detected and found to have infected 35 people in China [16]. There is also recent genomic evidence of a henipa-like Peixe-Boi virus in opossums in Brazil [17]. The global distribution and movement of henipavirus reservoir pteropid bats and the identification of new, related viruses and hosts make henipaviruses an ongoing threat to human and livestock health.

The structural protein M is highly conserved between HeV and NiV, sharing 89% and 96% amino acid sequence identity and similarity, respectively [18]. Henipavirus M is sufficient to direct the budding and release of virus-like particles [19], and in the context of the virus replication cycle, M promotes the efficient production of infectious virus particles [20]. Dimerization of M and subsequent interactions with the plasma membrane lipids phosphatidylserine and phosphatidylinositol 4,5-bisphosphate further promote M oligomerization, invoking conformational changes in the membrane curvature that promote virion formation and budding [21].

Whilst the budding roles of M occur in the cytoplasm, a transient localization of M to the nucleus is critical for its function. Discrete nuclear punctate localization of NiV-M has been observed at 8–16 h post-infection, after which NiV-M localizes to the plasma membrane [22,23]. The recombinant expression of M also recapitulates this phenomenon. A loss of the budding phenotype was observed when M was mutated such that it did not undergo nuclear–cytoplasmic trafficking [23,24,25,26]. Residue K258 plays a key role in M nuclear–cytoplasmic trafficking, with the monoubiquitination of K258 important for both trafficking and budding [23]. Cells with chemically depleted ubiquitin pools resulted in nuclear retention and failure to bud, as did K258A and K258R mutants [23,24,25]. Analysis of ubiquitination of lysine residues revealed that K258 is critically monoubiquitinated, supporting the hypothesis that ubiquitination of K258 is required for either a conformational change of M or downstream interaction with a ubiquitin-binding protein. Ubiquitination-driven trafficking of other paramyxovirus M proteins (Sendai virus-M, mumps virus M, but not Newcastle disease virus (NDV-M) or measles virus M) has also been demonstrated in subsequent studies [25].

The best-characterized method of nuclear import is the classical importin alpha (IMPα)/importin beta (IMPβ)-mediated pathway, a highly regulated process that viruses often manipulate to gain access to the nucleus. In this pathway, proteins generally display a classical NLS consisting of either one (monopartite) or two (bipartite) clusters of basic residues that bind with high affinity to the acidic pockets on the concave surface of the adaptor protein IMPα. These binding sites on IMPα are denoted the major site, located within armadillo repeat motifs (ARMS) 2–4, and the minor site, located within ARMS 7–8. Cargo-bound IMPα interacts with IMPβ via the IMPβ binding domain (IBB), and IMPβ mediates translocation through the nuclear pore in an energy-dependent manner [27,28,29,30].

The structural basis for NLS-IMPα binding was first described by Fontes, Teh, and Kobe [31]. Two prototypical NLSs were described: a monopartite NLS of one positive patch of amino acids found in simian virus 40 large T-antigen and a bipartite NLS containing two positive patches separated by a linker found in nucleoplasmin. Thereafter, numerous structures have been identified on the interface between viral proteins and IMPα, including HeV and NiV W protein [32], influenza A NP [33] and PB2 [34], Epstein-Barr virus NA-LP [35], Zika and Dengue 2 NS5 [36,37], and MERS ORF4B [38].

Consistent with nuclear import via the classical IMPα–IMPβ pathway, M can be coprecipitated with the human IMPα isoforms 1–7 [25]. Furthermore, a nuclear export signal (NES) for M has been defined, and studies have identified two regions within M that affect nuclear import [23]; a putative classical monopartite NLS (henceforth referred to as NLS1) corresponding to residues ^82^KRKKIR^87^ in both HeV and NiV, and a putative bipartite NLS (henceforth referred to as NLS2) corresponding to residues ^244^RRAGKYYSVEYCKRK^258^ in HeV M and ^244^RRAGKYYSVDYCRRK^258^ in NiV M. The crystal structures of HeV-M [39] and NiV-M [21] reveal that both NLSs are exposed and solvent accessible, with NLS1 positioned on a flexible loop and NLS2 within an α-helix (Figure 1). Mutations within NLS2 significantly impair M nuclear import and budding, while ubiquitination of M at a specific lysine residue (K258 in NiV M) in NLS2 regulates M nuclear–cytoplasmic trafficking. A mutation of NLS1 was noted to yield variable localization and less dramatic phenotypes than NLS2, but was not studied further [23].

Here, we sought to define the structural basis for M interaction with IMPα and to further assess the potential contribution of NLS1 to M nuclear import. We demonstrate that both M NLS regions can engage IMPα, the NLS1 at the major binding site, and the NLS2 at the minor site. We used recombinant expression and structure-guided mutagenesis of the NLSs to determine the phenotypic changes in nuclear localization and IMPα interaction. We confirm that the K258A mutation in NLS2 results in a loss of interaction with representative IMPα isoforms across the three IMPα subfamilies. However, this NLS2 mutation did not abolish M nuclear localization. A significant change in M nuclear localization was only observed following mutagenesis of NLS1 and NLS2 in the context of an NES mutant, suggesting that both regions contribute to NiV M nuclear localization.

## 2. Materials and Methods

### 2.1. Plasmids

Mouse IMPα1 (Uniprot accession P52293) residues 70–529 (lacking the IMPβ binding domain) was cloned into pET30 vector with an N-terminal 6His tag [31]. The genes for human IMPα proteins lacking the IMPβ binding domain (IMPα1 aa 71–529, IMPα3 aa 64–521, and IMPα5 aa 74–538, Uniprot accession P52292, O00629, and P52294, respectively) were codon optimized for *Escherichia coli* expression and cloned into pET30a(+) vectors at a BamHI restriction site with a TEV protease cleavage site (ENLYFQS) (Genscript).

The sequences for wildtype and mutant NiV M (Uniprot accession Q9IK90) aa 1–352 were cloned with N-terminal Flag tags into mammalian expression plasmid pCAGGS, as previously described [19]. NiV M mutants K84A, K84E, R256A, R257A, R82A/K83A/K84A, R256A/R257A, and RRK256/257/258AAA were generated by overlapping PCR. HA-tagged IMPα1, 3, and 5 have previously been described [40].

### 2.2. Protein Expression and Protein Purification

Plasmids were transformed into pLysS BL21 *E. coli* cells. Human IMPα1 was expressed using the IPTG method, as previously described [41], whereas all other IMPαs were expressed via auto induction [42]. Cells were harvested via centrifugation and resuspended in buffer A (50 mM phosphate buffer (pH 8.0), 300 mM sodium chloride, 20 mM imidazole), and frozen for future use. Cell lysis was initiated with freeze–thaw cycles followed by the addition of lysozyme (20 mg/mL) and DNase (5 μg/mL) to reduce viscosity. The soluble extract was obtained by centrifugation at 12,000 rpm, and protein purification was performed on a pre-equilibrated Ni-NTA affinity column in buffer A, followed by a 5 column volume wash with buffer A, then elution with buffer B (50 mM phosphate buffer (pH 8.0), 300 mM sodium chloride, 500 mM imidazole) over a 5 column volume gradient. Proteins were further purified via gel filtration on a Superdex 200 pg 26/600 column (GE Healthcare), equilibrated in a buffer containing 50 mM Tris (pH 8.0) and 125 mM sodium chloride. Fractions were assessed using SDS–PAGE and concentrated for subsequent experiments.

### 2.3. NLS Peptides

Peptides for the M NLS sequences were derived from the HeV isolate Horse/Autralia/Hendra/1994 (Uniprot identifier O89341) and synthesized by Genscript. The NLS1 peptide is comprised of residues HeV/NiV ^80^SGKRKKIRTIAAYPLGVGKS^99^ and NLS2 HeV ^243^VRRAGKYYSVEYCKRKID^260^. Peptides for structural studies were untagged, while peptides for binding studies were tagged with an N-terminal FITC label attached to an AHX linker. The lyophilized peptides were resuspended in 50 mM Tris (pH 8.0), 125 mM sodium chloride buffer to 10 mg/mL or 5 mg/mL for untagged and FITC conjugated peptides, respectively.

### 2.4. Crystallization and Data Processing

The purified IMPα proteins were each mixed with the untagged HeV-M NLS peptides at a molar ratio of 1:2 (IMPα:peptide) and screened in 48 well plates using the hanging-drop vapor diffusion method. Single crystals were harvested and transferred in cryoprotectant, cryo-cooled in liquid nitrogen, then shipped to the Australian Synchrotron. X-ray diffraction data were collected on the Macromolecular Crystallography MX2 beamline and Eiger detector [43]. CCP4 suite software was used to process the data with indexing and integration being performed in *iMosflm* [44]. The data were merged and scaled in Aimless [45], structures were solved by molecular replacement by Phaser [46], and rebuilding and refinement were performed in Phenix [47,48] and Coot [49], respectively. Finalized structures were deposited and validated within the Protein Data Bank, and interfaces analyzed by PDBSUM [50] and PISA Server [51].

### 2.5. Fluorescence Polarization

FITC-tagged HeV-M NLS peptides (10 nM) were incubated with two-fold serially diluted IMPα concentrations (starting concentration 200 µM) across 23 wells to a total of 200 µL. Fluorescence polarization measurements were performed on a CLARIOstar Plus plate reader (BMG Labtech, Ortenberg, Germany) in 96-well black Fluotrac microplates (Greiner Bio-One; Kremsmünster, Austria). Assays were performed in triplicate and contained a negative control (no binding partner). Triplicate data were plotted to a single binding curve using GraphPad Prism 9.5.1.

### 2.6. EMSA

Ten μL of 0.5 mg/mL FITC labelled M NLS peptides were combined with 40 μM of each IMPα isoform and incubated in a total volume of 15 μL for 15 min at room temperature. Three microliters of 50% glycerol were added, and samples were run on a 1.5% agarose TB gel for 1.5 h at 70 V in TB running buffer. The gel was stained using Coomassie stain and then destained in 10% ethanol and 10% glacial acetic acid. The images were recorded using a Gel Doc BioRad Gel Doc imaging system.

### 2.7. Co-Immunoprecipitation (Co-IP) Assay

HEK293T cells (1 × 10^6^ cells/well) were transfected with 2 μg Flag-NiV-M and 2 μg HA- IMPα or empty pCAGGS using lipofectamine2000 (Thermofisher Scientific, Waltham, MA, USA). Cells were lysed in 0.5% NP-40 lysis buffer (50 mM Tris pH 7.5, 280 mM NaCl, 0.5% NP-40, 0.2 mM EDTA, 2 mM EGTA, 10% glycerol, protease inhibitor (complete; Roche, Indianapolis, IN, USA). To precipitate HA- IMPα, cell lysates were incubated with EZview Anti-HA agarose affinity gel (Sigma-Aldrich, St. Louis, MO, USA) for 1 h at 4 °C with rocking. After 1 h, beads were washed 4 times with NP-40 lysis buffer and then eluted with HA peptide (Sigma-Aldrich) for 30 min at 4 °C with rocking. Whole cell lysates and co-IP samples were analyzed by western blot.

### 2.8. Western Blot

Lysates were run on a 10% Bis Tris plus polyacrylamide gel (Thermofisher Scientific, Waltham, MA, USA) and then transferred to a PVDF membrane (Sigma-Aldrich). Membranes were blocked in 5% non-fat dried milk in 1X phosphate buffered saline with 0.1% Tween20 (PBST) and probed with the indicated antibodies. Blots were developed using Western Lighting Plus ECL (Perkin Elmer) and imaged on ChemiDoc MP Imaging System (Bio-Rad). Antibodies: rabbit anti-flag (Sigma-Aldrich, cat #F7425), rabbit anti-HA (Invitrogen, Waltham, MA, USA, cat #71-5500), mouse anti-β-tubulin (Sigma-Aldrich, cat# T8328).

### 2.9. Immunofluorescence Assays

Huh7 cells (3 × 10^4^ cells/well) were seeded on glass coverslips. Cells were transfected with 500 ng of flag-tagged wildtype and mutant NiV-M plasmids with lipofectamine2000 (Thermofisher Scientific). Twenty-four hours post transfection, cells were fixed with 4% paraformaldehyde and permeabilized with 0.1% Triton X-100. Cells were stained with Flag-tagged monoclonal antibody conjugated to Alexa Fluor 488 (Invitrogen, cat# MA1-142-A488) and Hoechst 33342, trihydrocholoride trihydrate (Invitrogen, cat #H3570). The Biotek (Winooski, VT) Cytation 5 was used to calculate the average nuclear fluorescence over the average cytoplasmic fluorescence (Fn/c). Images of the same coverslips were taken with a Zeiss LSM 800 confocal laser scanning microscope at ×64.

## 3. Results

### 3.1. Henipavirus M NLS1 Binds to IMPα at the Major Site

Since M has been shown to engage the IMPα subunits via two NLSs (Figure 1) [25], we sought to elucidate the interface of both M NLSs with IMPα at high resolution using X-ray crystallography. We produced crystals containing the HeV M NLS1 bound to IMPα2. The structure of the complex was resolved to 1.9 Å, with crystals belonging to P2_1_2_1_2_1_ space group, and a single complex of M NLS1:IMPα2 present in the asymmetric unit. The M NLS1 was bound in the major binding site of IMPα2 (ARMS 2–4) (Figure 2A), with HeV M residues ^81^GKRKKIR^87^ interacting with IMPα2 through 14 hydrogen bonds, 1 salt bridge, and 131 non-bonded contacts [50,51] (Figure 2B,C). HeV M residues ^84^KKIR^87^ occupied the P2-P5 sites of IMPα2, respectively, with K84 making interactions at the IMPα2 P2 site, involving Gly150, Thr155, and Asp192 (Figure 2B,C). The interaction interface extended over 895 Å^2^ of surface area [51]. The minor binding site of IMPα2 was also occupied by electron density that corresponds to HeV M NLS1 residues *^81^*GKRK*^84^*; however, the binding of excess peptide in the minor site is a common and well-described phenomenon in the literature [31,52]. Moreover, this binding was not observed in the IMPα3 structure (see below) (see Table 1 for complete collection and refinement statistics).

The structure of IMPα3 bound to HeV M NLS1 was also determined and resolved to 2.75 Å in the space group P2_1_2_1_2_1_ (but a distinct crystal form to the IMPα2:M NLS1 crystal described above; see Table 1 for unit cell comparison). The asymmetric unit was comprised of a single complex of the M NLS1 bound to IMPα3, and the M NLS1 was bound in the major binding site of IMPα3 in a similar fashion to IMPα2 (see Appendix A). In this structure, HeV M ^84^KKIR^87^ occupied the P2–P5 sites of IMPα3, formed 14 hydrogen bonds, 1 salt bridge [51], 99 non-bonded contacts [50], and exhibited a buried surface area of 812 Å^2^. Similar to the IMPα2 interface, K84 also formed interactions at the IMPα3 P2 site, involving Gly145, Thr150, and Asp187 (see Table 1 for complete collection and refinement statistics). Since the residues bound at the HeV NLS were identical in the NiV sequence, we did not solve the structure of the NiV NLS1 bound to IMPα.

### 3.2. M NLS2 Binds to IMPα at the Minor Site

We also used X-ray crystallography to determine the binding interface of HeV M NLS2 bound with IMPα. A structure was resolved at 2.1 Å, with the complex of HeV M NLS2 bound to IMPα2 in the P2_1_2_1_2 space group (the same space group and crystal form as M NLS1:IMPα2). Here, we found the NLS2 peptide bound in the minor binding site of IMPα2 (ARMS 6–8) (Figure 2D). Whilst density at the major site of IMPα2 was present, it did not correspond to the HeV M NLS2 sequence, and subsequent analysis of IMPα2 crystals by mass spec analysis revealed this to be an E. coli contaminant (30s ribosomal protein) corresponding to KKLARE. Five residues of the HeV M NLS2 could be reliably modelled at the minor site, with the main chain and side chains well resolved, and revealing ^255^CKRKI^259^ at the interface, with K256, R257, and K258 bound at the P1′–P3′ sites, respectively (Figure 2E,F). The interaction was mediated through eight hydrogen bonds, 2 salt bridges, 68 non-bonded contacts [50,51], and a buried surface area of 361.1 Å^2^ (Figure 2E,F; see also Table 1 for a full collection of data collection and refinement statistics).

### 3.3. M NLS1 and NLS2 Bind IMPα Isoforms in Electromobility Shift Assays and Fluorescence Polarization Assays

Based on our structural data indicating that both NLS1 and NLS2 of M could bind IMPα, we assessed whether these NLSs may exhibit specificity towards IMPα isoforms as observed with other viral cargo, such as the NIV and HeV W proteins [53,54]. Firstly, we performed an agarose electromobility shift assay with recombinant IMPα proteins from each of three subfamilies (α1: IMPα1, α2: IMPα3, and α3: IMPα5) with the HeV M NLS peptides. In this assay, IMPα proteins migrated toward the anode (Figure 2G, lane 3–5), while the HeV M NLS peptides migrated toward the cathode (Figure 2G, lane 1 and 2). When combined with IMPα1, IMPα3, and IMPα5, the migration of FITC-labelled NLS1 peptides shifted towards the anode, but different to that of unbound IMPα proteins, indicating binding (Figure 2G, lanes 6–8). When combined with IMPα1, IMPα3, and IMPα5, the NLS2 peptide also shifted migration towards the anode, indicating binding (Figure 2G, lanes 9–11). These results are consistent with the crystal structures, suggesting that both NLS1 and NLS2 peptides can form a stable complex with IMPα isoforms across all three subfamilies.

To quantitatively assess these interactions, we employed fluorescence polarization to measure the binding affinity of the HeV M NLS peptides with IMPα isoforms. We observed binding of NLS1 with IMPα1, IMPα3, and IMPα5 with dissociation constants (K_D_) in the range of 0.56–2.57 μM (Figure 2H). We observed similar binding of the NLS2 with IMPα1, IMPα3, and IMPα5, ranging between 5.28–11.54 μM (Figure 2I). These results were consistent with both the crystal structure data and the electromobility shift assay.

### 3.4. The M NLS2 Is an Important Binding Interface in a Cellular Context

With the in vitro data indicating both NLS1 and NLS2 of M have the capacity to form a stable interaction with IMPα, we designed specific, structure-guided point mutations to examine the M-IMPα interactions and assessed these in a cellular context using the full-length proteins by cco-IP. Here, we found that the full-length M protein of NiV was able to co-IP with members of the three IMPα subfamilies, IMPα1, 3, and 5, consistent with a prior report (Figure 3A–F) [25]. Based on our structural data for NLS1, we mutated the Lys84 residue, positioned at the critical P2 site of IMPα [36,54,55] to either Ala (M K84A) or a charge-reversed Glu (M K84E). Neither mutation impaired the co-IP of M with IMPα1, IMPα3, and IMPα5 (Figure 3A–C). We therefore introduced a more drastic mutation, ^83^RKK to ^83^AAA (M R83A/K84A/K85A), and also found that this did not impair the binding in these assays. The results suggest that NLS1 is not required for interaction with IMPα in co-IP experiments. We next assessed the impact of mutations within NLS2 on NiV-M interaction with IMPα by co-IP (Figure 3D–F). Individual point mutations at R256A P1′ and R257A P2′ (each mediating 3 and 4 H-bonds respectively) did not affect binding with IMPα1, IMPα3, and IMPα5; however, mutation of K258A, mediating 3 H-bonds with IMPα residues G281, N283, and T322 at the P3′ site abrogated binding. Double mutations of NLS2 ^256^RR to ^256^AA (M R256A/R257A) or a triple mutation of ^256^RRK^258^ to ^256^AAA^258^ (M R256A/R57A/K258A) resulted in severely reduced protein expression (Figure 3D–F). Therefore, it was not possible to assess the combined contribution of these residues to IMPα binding in cell-based assays. Cumulatively, these data demonstrate that NLS2 residue K^258^ plays an important role in M-IMPα interaction, as detected by co-IP.

To determine the phenotypic changes of these mutations on nuclear transport of M, we performed indirect immunofluorescence assays (IFA) using Huh7 cells and recombinant Flag-NiV-M. Because the nuclear export to the cytoplasm could potentially confound interpretation, IFA was performed using a NiV-M with a mutant NES (L106A, L107A, and L110A). Distinct nuclear localization was observed following mutation of the NES for WT NiV-M, demonstrating successful inhibition of NiV-M nuclear export (Figure 4A). When mutations were introduced in either NLS1 (M R82A/K83A/K84A) or NLS2 (M K258A) in the context of the NES mutant, M localization remained predominantly nuclear, suggesting that nuclear import was still occurring (Figure 4A). However, when mutations were introduced within both NLS1 and NLS2, M localization was severely mislocalized to the cytoplasm (Figure 4A). Calculating the average nuclear fluorescence over the average cytoplasmic fluorescence (Fn/c) demonstrated that combined mutations in both NLS1 and NLS2 in the NES mutant background led to a significant cytoplasmic shift in M localization (Figure 4B). Co-IP experiments confirmed that the NSL1 and NLS2 combined mutant, such as the NLS2 (M K258A) mutant, did not co-IP with IMPα1 (Figure 4C). While the co-IP experiments suggest that K^258^ in NLS2 is critical for M:IMPα interaction, the IFA data demonstrate that the nuclear localization of M is only impaired when both NLS1 and NLS2 are mutated. This suggests that NLS1 also contributes to M nuclear import in a manner that cannot be fully appreciated in cIP experiments alone.

## 4. Discussion

Viruses employ many mechanisms to hijack cellular processes to promote their replication, pathogenicity, and innate immune evasion. Despite replicating in the cytoplasm, RNA viruses target specific proteins to the nucleus, affecting normal cellular function. Many clinically significant species exploit this process, including HeV and NiV W proteins [53], influenza A NP [33], and PB2 [34], Epstein-Barr virus NA-LP [35], Zika and Dengue 2 NS5 [36,37], and MERS ORF4B [38]. The henipavirus M protein is another example of a viral protein that localizes to the nucleus both as a recombinant protein [23,24,25,26,56,57,58] and in the viral context [22,23,26]. Henipavirus M is a structural protein that requires nuclear import for its ubiquitination, assembly, and membrane targeting. M nuclear import is also important for accessory functions involved in the modulation of host cell pathways [56,57,59]. The data presented here report the non-classical mechanisms by which M traffics to the nucleus and provides insight into the varied ways that viruses hijack host cellular pathways.

While our data demonstrate that M utilizes the classical IMPα/IMPβ-mediated pathway, the best-characterized nuclear import pathway, our data also demonstrate that henipavirus M NLS2 employs a non-classical binding mechanism. Specifically, M residues ^255^CKRKI^259^ interact with only the IMPα minor binding site. In contrast, prototypical classical NLSs bind to IMPα through occupying either the major site only, between ARMs 2–4, or adopt an extended conformation through both the major and minor sites as a bipartite NLS, through ARMS 2–4 and 7–8 [31,60]. The IMPα minor binding site is defined as P2′-P5′ with the critical P2′ site IMPα residues E^396^ and S^360^, and P3′ site IMPα residues G^281^, N^283^, and T^322^ [61]. Here, M NLS2 R^257^ and K^258^ residues occupy P2′ and P3′ forming 4 H-bonds and 3 H-bonds, respectively, and this binding conformation has been observed in other cargo binding the minor site. Similar to our structure, Lott [62] described the minor site-only NLS of human phospholipid scramblase 4 binding to IMPα in the same antiparallel conformation as M NLS2, with the P2′ and P3′ sites occupied by an R and K, respectively. Viral proteins have also been shown to employ this binding mechanism, with the non-classical NLS of influenza A virus nucleoprotein binding to the IMPα minor site through an RK motif [63].

An additional notable feature of the M NLS2-IMPα interaction detailed in this study was that the NLS2 peptide ^244^RRAGKYYSVEYCKRK^258^ did not bind ^244^RR^245^ residues that have been reported to be part of a functional bipartite NLS [23,24,25,26]. Our structure revealed that the CKRK^258^ residues bound at the minor site, precluding the ability of NLS2 to bind as a bipartite since NLS cargo bind IMPα in an antiparallel manner. This requires the N-terminus of a bipartite NLS to bind at the minor site, and the C-terminal basic cluster to bind at the major site. We also note that the NLS2 region undergoes a conformational change upon binding IMPα, since region 252–261 is helical in the full-length context [21,39], but linear in our structure. This is consistent with other IMPα cargo that have been shown to undergo a conformational change upon IMPα binding (e.g., chloride intracellular channel protein 4 (CLIC4) NLS and Tick-borne encephalitis capsid protein (TBEVC)) [64,65].

The structure of NLS2:IMPα is consistent with our co-IP findings that NLS2 is critical for interaction with IMPα (Figure 3D–F). Wildtype M co-IPs with IMPα across each of the three subfamilies. This is also consistent with our gel shift assay, with both NLS peptides binding IMPα1, 3, and 5. This is further supported by proteomics and co-IP by Pentecost et al. [25]. Due to severely reduced expression of the NLS2 R256A/R257A mutants in the present study, the contribution of these residues to co-IP with IMPα could not be determined. Here, structure-guided point mutations of NiV M R256 (HeV-M K256 equivalent) and R257 failed to abrogate co-IP with IMPαs, while M K258 was shown to be an important for binding, as determined by co-IP interaction, with K258A completely abolishing the interaction with IMPα 1, 3, and 5. Although the P2′ residue R257 is solvent exposed and contributes three H-bonds and two salt bridges to the interface, it appears that the P3′ residue, K258, is the major determinant of binding. We hypothesize that this phenomenon may be due to a switch in topology, driven by K258, that activates an IMPα specific NLS. Thus, it is conceivable that the point mutation K258A prevents fold switching of the α8 helix and prevents IMPα binding, as seen in the co-IP. A fold-switch may not be prevented by the point mutations R256A/R257A and K258R (not shown), allowing mutant NiV-M proteins to engage IMPα in the context of co-IP. A fold-switch within this region of the protein also supports the multifunctionality of the K258 residue. The K258A mutation prevents HeV M localization to the subnucleolar compartments [26], interaction with nucleolar protein treacle [26,59], decreased nuclear export, decreased budding [23,24,25,26,59], and loss of IFN-I antagonism [66]. Furthermore, a conformational change of the α8 helix may be required to accommodate monoubiquitination at K258 [23].

In addition to NLS2, we examined the largely neglected putative monopartite NLS, ^81^GKRKKIRTI^89^. Here, X-ray crystallography revealed that NLS1 engages IMPα as a classical monopartite NLS. Lys84 was shown to be positioned within the P2 site, forming interactions with the well characterized IMPα Gly150, Thr155, Asp192, and the binding was consistent with other NLS cargo binding at the major site, including SV40 T-ag [31] and HIV Tat [67]. Interestingly, the NLS1 region is exposed and accessible in the full-length M structure [21,39]. Whilst mutagenesis of NLS1 was insufficient to disrupt co-immunoprecipitation with IMPα, our observation that only when both NLS1 and NLS2 binding regions are mutated does M protein mislocalize to the cytoplasm, points to a role for NLS1 in M nuclear import. It is possible that NLS1 and NLS2 may either cooperate to form an enhanced interaction with IMPα, or may be utilized at different stages of the virus lifecycle (for example, M oligomerization or post-translational modifications, such as lysine acetylation [23], ubiquitination [23,25], or NLS phosphorylation, may switch NLS usage) [36,68]. This is supported by these regions binding IMPα at different sites (NLS1 at the major site, NLS2 at the minor site). The finding that none of the mutations tested completely abrogate nuclear localization of M could also suggest the possibility that additional or multiple nuclear import pathways beyond IMPα/IMPβ are used. This possibility is supported by NiV and HeV M interactions with a number of nuclear trafficking and nuclear pore proteins [25]. In addition, the paramyxovirus, NDV, has a similar M structure [69] and is reported to be imported to the nucleus via IMPβ, independent of IMPα [70]. Future investigation into the nuances of HeV and NiV M nuclear import pathways could incorporate the use of specific nuclear receptor inhibitors.

In addition to budding and release of viral particles, henipavirus M has other nuclear functions. A genome-wide siRNA screen of host cell factors required for HeV infection identified 43 genes that significantly reduce viral titers when silenced. Fibrillarin (a nucleolar methyltransferase) led to the most significant reduction of 99.9% and was found to co-localize with M in the nucleus. Out of the 66 genes of mid-high confidence that affected henipavirus infection, approximately a third localize to the nuclear compartments, including seven out of the ten that have the most significant impact on the virus [56]. Furthermore, the CRM-1 nuclear export adapter protein ANP32B has been identified as an M interactor. The proteins co-localize in the nucleus both in recombinant context and in viral infection, and ANP32B copurifies with M in cells expressing both [57]. Rawlinson et al. described the specific activation of the nucleolar protein treacle by M [26,59]. They report that while HeV infection results in DNA damage, M protein interacts with treacle and inhibits ribosomal RNA synthesis as part of the DNA damage response. This process occurs within the nucleus, specifically the nucleolus, of cells. Together, these studies underscore the significant and complex role of henipavirus M protein nuclear trafficking during the henipavirus life cycle and demonstrate the importance of understanding the mechanisms that govern M nuclear import.

## Figures and Tables

**Figure 1 viruses-15-01302-f001:**
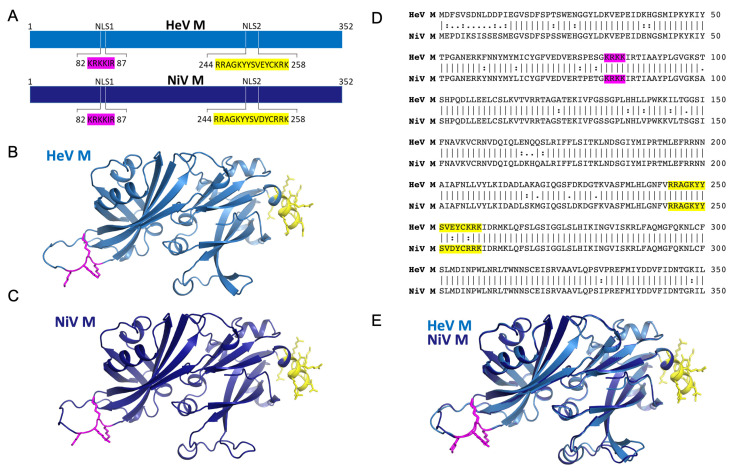
Henipavirus matrix protein possess two putative NLS regions. (**A**) Schematic showing the putative NLS regions of HeV M and NiV M. (**B**) Structure of HeV M in cartoon representation (blue) with the putative monopartite NLS (magenta) and bipartite NLS (yellow) highlighted. The structure was created using alphafold (template PDB70) due to missing loops in the structure of PDB 6BK6. (**C**) Structure of NiV M in cartoon representation (dark blue) with the putative monopartite NLS (magenta) and bipartite NLS (yellow) highlighted. The structure was created using alphafold (template PDB70) due to missing loops in the structure of PDB 7SKT. (**D**) Pairwise sequence alignment of HeV and NiV M proteins (**E**) Pymol image showing superposition of HeV/NiV M from B and C respectively. The RMSD of PDB 6BK6 (HeV M) and 7SKT (NiV M) is 0.57 Å.

**Figure 2 viruses-15-01302-f002:**
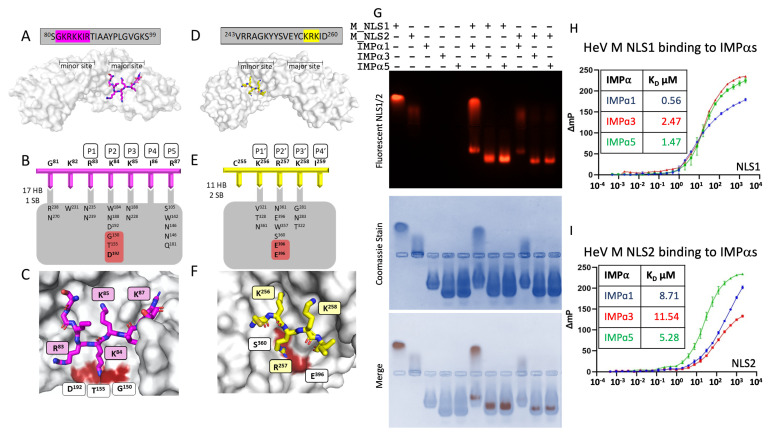
Structural basis for the interaction between the matrix putative NLS sequences and IMPα. (**A**) HeV M NLS1 peptide sequence displayed in grey box with resolved amino acids highlighted in magenta. Matrix NLS1 (magenta sticks) binds to IMPα (grey surface) at the major site. (**B**) Schematic of the interface between NLS1 (magenta) and IMPα (grey). HeV M K^84^ binds to P2 site on IMPα (Gly150, Thr155, and Asp192) (highlighted red) with H-bonds and salt bridge interactions depicted. (**C**) The resolved amino acids of HeV M NLS1 (magenta sticks) are shown in the major binding site of IMPα. (**D**) HeV M NLS2 peptide sequence displayed in the grey box with resolved amino acids highlighted in yellow. Matrix NLS2 (yellow sticks) binds to IMPα (grey surface) at the minor site. (**E**) NLS2 binds to IMPα as a non-classical, minor-site only NLS. Schematic of the interface between NLS2 (yellow) and IMPα (grey). Hydrogen bonds are shown in standard font, and salt bridges in bold font. (**F**) The resolved amino acids of HeV M NLS2 (yellow sticks) are shown in the minor binding site of IMPα, with R^257^ forming salt bridges with Glu^396^ and Ser^257^ (highlighted in red). (**G**) Electromobility shift assay (EMSA) showing NLS1 in lane 1, NLS2 in lane 2, IMPα1, 3, and 5 in lanes 3, 4, and 5, respectively. NLS1 combined with IMPα1, 3, and 5 in lanes 6–8, respectively, and NLS2 combined with IMPα1, 3, and 5 in lanes 9–11, respectively. (**H**,**I**) Fluorescence polarization assays measuring binding strength between NLS1 and NLS2 of HeV and NiV M proteins respectively.

**Figure 3 viruses-15-01302-f003:**
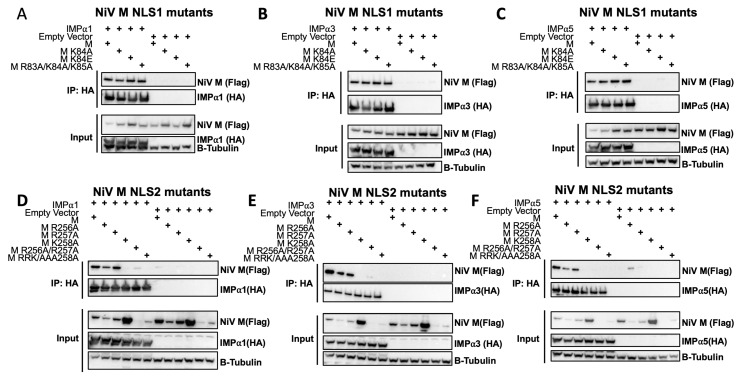
Recombinant NiV M engages human importin alpha subunits 1 and 3 and undergoes nuclear import. Co-immunoprecipitation assays were preformed examining the interaction between NiV M NLS1 mutants (K84A, K84E, and R83A/K84A/K85A) (**A**–**C**) or NiV M NLS2 mutants (R256A, R257A, K258A, R256A/R257A, and R256A/R257A/K258A) (**D**–**F**). Assays were performed be immunoprecipitating HA-tagged IMPα1 (**A**,**D**), IMPα3 (**B**,**E**), and IMPα5 (**C**,**F**) isoforms and were probed for the presence of Flag-tagged NiV M. Expression of β-tubulin is used as a loading control for the input cell extracts.

**Figure 4 viruses-15-01302-f004:**
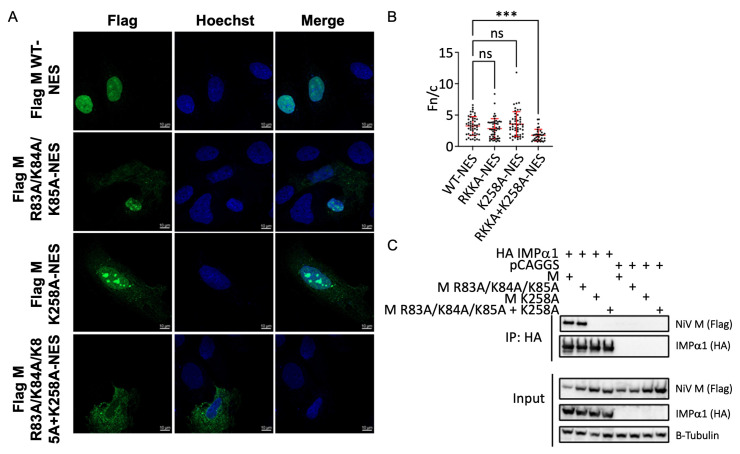
NiV M NLS1 and NLS2 contribute to NiV M nuclear import. (**A**) Confocal microscopy images of Huh7 cells transfected with flag-tagged NiV M-NES plasmids. Following transfection, cells were stained with Hoechst 33,342 and an anti-Flag antibody conjugated to AlexaFluor488. Images were captured using a Zeiss LSM 800 confocal microscope. (**B**) Quantitation of average nuclear fluorescence over average cytoplasmic fluorescence (Fn/c). Fn/c was calculated using the Biotek Cytation 5. Data are represented as mean fn/c for each sample ± SD. NS denotes *p* value > 0.05. *** denotes *p* values ≤ 0.001. (**C**) Co-IP assay measuring the interaction between HA-tagged IMPα1 and flag-tagged NiV M mutants (R84A/R84A/R85A, K258A, and R84A/R84A/R85A + K258A). Assays were performed as in Figure 3A–F.

**Table 1 viruses-15-01302-t001:** Data collection and refinement statistics for crystal structures of IMPα and HeV M NLS1 and NLS2.

Data Collection and Processing	IMPα2:HeV M NLS1	IMPα2:HeV M NLS2	IMPα3:HeV M NLS1
Wavelength (Å)	0.9537	0.9537	0.9537
Resolution range (Å)	24.42–1.9 (1.94–1.9)	19.78–2.10 (2.16–2.10)	29.78–2.75 (2.9–2.75)
Space group	P 21 21 21	P 21 21 21	P 21 21 21
Unit cell (Å, ^o^)	78.49 89.86 99.83 90 90 90	78.08 89.50 97.06, 90 90 90	49.06 64.27 158.58 90 90 90
Total reflections	405,486 (26,486)	167,351 (12,946)	154,985 (23,139)
Unique reflections	56,298 (3749)	40,357 (3253)	13,681 (1966)
Multiplicity	7.2 (7.1)	4.1 (4.0)	(11.8)
Completeness (%)	99.9 (100)	99.9 (99.9)	99.9 (100)
Mean I/sigma (I)	11.9 (1.5)	11.0 (1.9)	7.8 (2.3)
Wilson B-factor Å^2^	29.65	35.12	55.58
R-merge	0.085 (1.376)	0.063 (0.696)	0.181 (1.032)
R-pim	0.050 (0.820)	0.035 (0.400)	0.078 (0.446)
CC_1/2_	0.998 (0.602	0.998 (0.734)	0.995 (0.887)
Refinement			
Number of reflections	56,237 (5567)	40,289 (3941)	13,635 (1323)
Number of R-free reflections	2864 (322))	1981(180)	671 (80)
R-work %	0.1728 (0.2853)	0.1791 (0.2520)	0.2411 (0.3185)
R-free %	0.1900 (0.3147)	0.2053 (0.3058)	0.2811 (0.3553)
RMS (bonds)	0.015	0.006	0.003
RMS (angles)	1.18	0.77	0.58
Ramachandran			
favored (%)	98.6	98.14	97.85
allowed (%)	1.4	1.86	2.15
outliers (%)	0.00	0.00	0.00
Average B-factor Å^2^	43.57	50.33	72.53
Clash score	1.54	4.07	4.92
PDB code	8FUA	8FUC	8FUB

## Data Availability

Structures have been deposited to the Protein Data Bank (PDB codes 8FUA, 8FUC, 8FUB), and validated and released prior to manuscript submission.

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
