# Peer review of "Henipavirus Matrix Protein Employs a Non-Classical Nuclear Localization Signal Binding Mechanism"

_viruses, 2023, doi:10.3390/v15061302_

Round 1

Reviewer 1 Report

In this Manuscript, the authors study the structural basis of nuclear localization of Henipaviral matrix protein. Henipavirus is primarily harbored in bats and have shown zoonotic transmission leading to significant mortality with evidence of human transmission. The authors study the viral M protein that is important for virus budding during viral replication cycle. The authors did a good job in solving the structures of both NLS1 and NLS 2 peptide of M protein bound to importin a. NLS1 binds to the major site and NLS2 binds to the minor site on importin. Electrophoretic mobility shift assay and fluorescent polarization experiments show evidence of NLSs association with importin isoforms. They further went on to show that NLS2 binding is important for association in a cellular context, especially K258A mutant that did not co-immunoprecipitate with importin isoforms. However, cell staining suggests importance of both NLS1 and NLS2 for nuclear localization of the matrix protein. Based on the structural evidence, mutational analysis, cellular assay and coIP the authors show a non-classical way of localization of matrix protein to the nucleus where both NLSs are important.

Comment:

·      In structure IMPa2 with NLS1 (PDB 8FUA), there is density for NLS1 peptide in the minor site as well. This is absent with the other isoform (importin a3). The authors have failed to mention that and explain in the manuscript.

·      Have the authors looked at other technique like ITC to measure binding affinity to importin in context of WT and mutant full length M protein.

·      Why did the authors only try co-IP with Nipah virus M and not Hendra virus M?

·      I am intrigued to know if the authors have done co-IP with anti-FLAG antibody.  Proteomic analysis of this to identify additional binders of nuclear import would shed more light into the structure guided mutants that is supposed to disrupt interaction with importin a.

Author Response

  • In structure IMPa2 with NLS1 (PDB 8FUA), there is density for NLS1 peptide in the minor site as well. This is absent with the other isoform (importin a3). The authors have failed to mention that and explain in the manuscript.

We thank the reviewer for picking this up. The binding of excess peptide in the minor site is a common and well-described phenomenon in the literature. We have updated the manuscript to ensure this is clear. The manuscript now reads:

The minor binding site of IMPa2 was also occupied by electron density that corresponds to HeV M NLS1 residues 81GKRK84, however the binding of excess peptide in the minor site is a common and well-described phenomenon in the literature [31, 53]. Moreover, this binding was not observed in the IMPa3 structure (see below) (see Table 1 for complete collection and refinement statistics).

Comment

  • Have the authors looked at other technique like ITC to measure binding affinity to importin in context of WT and mutant full length M protein.

This is a good suggestion, however in vitro binding experiments of the full-length Matrix protein were not possible as the expression constructs were insoluble E. coli. Experiments examining the interactions within the full-length context (including mutants) were undertaken by co-IP.

Comment

  • Why did the authors only try co-IP with Nipah virus M and not Hendra virus M?

Given that the NLS1 of NiV and HeV are identical and that NLS2 are nearly identical between NiV and HeV, it seemed unnecessary to perform CoIPs with mutants from both NiV and HeV.

Comment

  • I am intrigued to know if the authors have done co-IP with anti-FLAG antibody.  Proteomic analysis of this to identify additional binders of nuclear import would shed more light into the structure guided mutants that is supposed to disrupt interaction with importin a.

We have not attempted to co-purify host cell interacting proteins of NiV or HeV M. However, other groups have performed such studies. In a study that included NiV and HeV M proteins, Pentecost et al. (PLoS Pathogens 2015 https://doi.org/10.1371/journal.ppat.1004739 <https://doi.org/10.1371/journal.ppat.1004739>) identified numerous importins and nucleoporins that co-precipitated with paramyxovirus M proteins. Martinez-Gil et al. (PloS Pathogens 2017 https://doi.org/10.1128/JVI.01461-17 <https://doi.org/10.1128/JVI.01461-17>) reported several co-precipitating host proteins. Of these, only Ran binding protein 2 (also known as Nup358) has an obvious connection to nuclear import.

Reviewer 2 Report

This is an interesting and important study that adds significantly to the field and will have a noticeable impact. The manuscript is well-written and concise. It reports a wealth of results generated by an imppressive set of the well-designed experiments. Data generated in this study provide string support to the conclusions of the manuscript.

However, in my view, this important work can be further enhanved by conducting an additional bioinformatics analysis of the full-length M proteins from HeV and NiV for their intrinsic disorder predispositions. This analysis in a form of the per-residue disorder profiles could provide some additional points for discussion. In fact, as it follows from my superficial analysis, where I utilized several of such tools, clearly indicated that although NLS1 motifs in HeV M and NiV M are located within relatively long disordered regions, the NLS2 motifs are placed within the "dual personality" regions, which are predicted as structured by some tools, but marked as disordered or flexible by others.       

Author Response

Reviewer 2

Comment

This is an interesting and important study that adds significantly to the field and will have a noticeable impact. The manuscript is well-written and concise. It reports a wealth of results generated by an imppressive set of the well-designed experiments. Data generated in this study provide string support to the conclusions of the manuscript.

Thank you!

However, in my view, this important work can be further enhanved by conducting an additional bioinformatics analysis of the full-length M proteins from HeV and NiV for their intrinsic disorder predispositions. This analysis in a form of the per-residue disorder profiles could provide some additional points for discussion. In fact, as it follows from my superficial analysis, where I utilized several of such tools, clearly indicated that although NLS1 motifs in HeV M and NiV M are located within relatively long disordered regions, the NLS2 motifs are placed within the "dual personality" regions, which are predicted as structured by some tools, but marked as disordered or flexible by others.

Without knowing the exact prediction software used by reviewer, it is difficult to replicate. We did perform a bioinformatics analysis of the Matrix disorder as suggested, and in general, found that NLS1 is predicted to be disordered, and NLS2 is predicted to be ordered (as the reviewer also found; see also a typical profile below). This is in agreement with the experimentally determined crystal structures, where the NLS1 could not be resolved due to disorder, while NLS2 contained some secondary structural elements. Therefore our preference would be to not include a discussion around NLS2 disorder as it’s not supported by the crystal structure, and there is ambiguity in the results generated by the prediction software.

Metapredict V2: An update to metapredict, a fast, accurate, and easy-to-use predictor of consensus disorder and structure
BioRxiv 2022.06.06.494887
Emenecker, R. J., Griffith, D., & Holehouse, A. S. (2022). Link to paper

ure has been adjusted accordingly.

Reviewer 3 Report

This paper reported the study of the interaction between henipavirus Matrix protein (M) nuclear localization signals (NLSs) and importin alpha (IMPα) with crystallography and biochemical validation assays. The interaction of both NLS peptides with IMPα was established, with NLS1 binding the IMPα major binding site, and NLS2 binding as a non-classical NLS to the minor site. These interactions were further confirmed by co-immunoprecipitation (co-IP) and immunofluorescence assays (IFA). This study demonstrated a supportive role for NLS1 in M nuclear localization and provided additional insight into the critical mechanisms of M nucleocytoplasmic transport, contributing the understanding of viral pathogenesis as well as a potential target for novel therapeutics for henipaviral diseases. This topic fits the scope of the journal, and the experiments can support the conclusions. In general, this manuscript was well-organized, and the references are updated. The key issues are required to be addressed before its publication on Viruses.

Major points:

1. In the fluorescence polarization assays, the binding affinities of NLS2 to HeV and NiV M proteins are fairly accepted, as the readout signals are still increasing with the increased concentrations of IMPα. The higher concentrations of IMPα in these assays are required to get the accurate Kd values.

Minor points:  

1. In Figures 2H and 2I, the font size in the tables (with Kd values) is required to be adjusted.

2. In Figure 3, the resolution and the label size are required to be adjusted.

The English language is acceptable.

Author Response

Reviewer 3

Comment

  1. In the fluorescence polarization assays, the binding affinities of NLS2 to HeV and NiV M proteins are fairly accepted, as the readout signals are still increasing with the increased concentrations of IMPα. The higher concentrations of IMPα in these assays are required to get the accurate Kd values.

We have performed the experiments again at higher concentrations of IMPA (previously 20 uM, now 200 uM). Higher concentrations of IMPA are not possible due to instability and autoinhibition.

Comment

Minor points:

  1. In Figures 2H and 2I, the font size in the tables (with Kd values) is required to be adjusted.

The font size has been increased.

  1. In Figure 3, the resolution and the label size are required to be adjusted.

The Figure has been adjusted accordingly.